# IMAGE-DRIVEN VIDEO EDITING WITH LATENT DIFFUSION MODELS

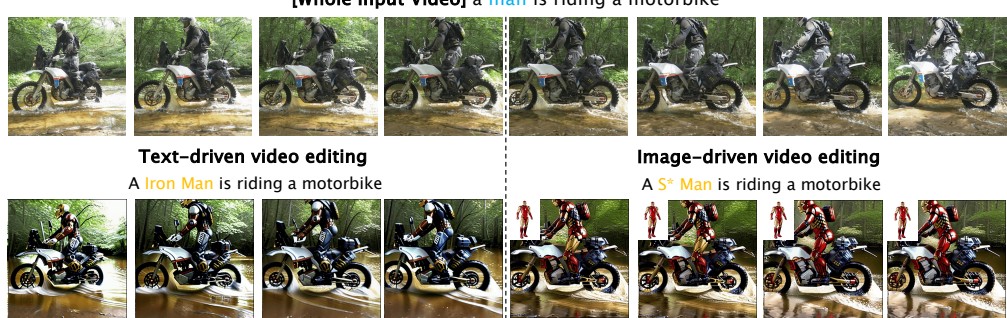

Figure 1: Illustration of the differences between text-driven video editing and the proposed image-driven video editing. The former modifies the video by changing the text prompt, as opposed to the proposal, which uses an image to provide detailed control over the editing process.

## ABSTRACT

Recent works in Text-to-Image (T2I) models have shown potential in addressing text-driven video editing using latent diffusion models (LDM). However, text prompts as a representation of visual signals remain a crude abstraction, leaving the challenge of achieving fine-grained and controllable video editing unresolved. In this study, we introduce the **L**atent prompt based **I**mage-driven **V**ideo **E**diting (LIVE) framework to unlock the capabilities of pretrained LDM for precise editing control. At the heart of LIVE lies a novel Latent Prompt Mechanism, which utilizes latent code from a reference image as a prompt to enrich visual details. We begin by revisiting the attention mechanisms in LDM and enhancing them to facilitate comprehensive interactions between video frames and latent prompts in both spatial and temporal dimensions. We also devise a training process to fine-tune components such as latent prompts, textual embeddings, and LDM parameters, effectively representing the provided video and image within the diffusion space. Subsequently, these optimized elements are combined to generate the edited video output, enabling seamless object substitution in each frame with user-specified targets while maintaining visual consistency across frames. Our experiments on real-world videos demonstrate the efficacy of the LIVE framework and its promising applications in image-driven video editing tasks.

## 1 INTRODUCTION

In recent years, significant progress has been made in video generation. Initially, the primary focus of video generation research (Ho et al., 2022c; Singer et al., 2022; Ho et al., 2022a; Harvey et al., 2022) is to generate a video from noise. However, with the emergence of AI Generated Content (AIGC) and rapid development of generative models like diffusion models (Ho et al., 2020) and GANs (Goodfellow et al., 2020), research emphasis has shifted towards video editing (Molad et al., 2023; Ceylan et al., 2023; Liu et al., 2023; Qi et al., 2023). Despite progress made in video editing, current video editing tasks are limited to text-based modifications, which fall short of providing the level of customization that users desire. To address this issue, we introduce an image-driven video editing task that allows users to change specific parts of a video based on user-input images.

We propose a solution to this task, which provides a more personalized and precise video editing experience that aligns with the user's intended modifications.

Currently, most methods utilize attention control (Hertz et al., 2022; Chefer et al., 2023) between text and images to manipulate and substitute objects or styles within video sequences, concurrently employing null-text inversion (Mokady et al., 2022) to uphold a consistent background. However, these existing methods have not yet explored the possibility of directly incorporating input images into the editing process. Our image-driven video editing task poses a unique challenge, involving the replacement of an object within an input image with a corresponding video component while preserving cross-frame consistency and adhering to the sequence's motion. This task differs from traditional video editing as it requires the utilization of both image and text inputs.

Taking inspiration from Tune-A-Video (TAV) (Wu et al., 2022), we design our image-driven video editing framework, based on the text-to-video (T2V) model, to address the image-driven edit task. Nevertheless, a newly built T2V model does not take into account the information contained within input images of uses. Thus, we divide our framework into two parts. The first part is fine-tuned on the original video, which focuses on learning the motion and background. The second part is trained to incorporate the input image into our model, achieving text and image alignment. After training these two parts together, during the inference process, we adopt attention control to maintain background consistency and generate final edit results.

In particular, our base model is implemented with inflated 2D stable diffusion, which has the ability to generate videos from text. For the first part, we modify the T2V framework with a learnable latent prompt to absorb motion information of input video. For the second part, we leverage textual inversion (Gal et al., 2022) which is a cost-effective method for aligning text and image to establish a mapping between text and image. We design a pipeline to train two parts together, improving both parts' training effects. At inference time, we replace the latent prompt and text with relevant training outcomes and utilize custom attention control to get edited video.

In summary, our contributions are in three folds: 1) We introduce a novel image-driven video editing task aimed at generating customized videos based on the input image; 2) We propose a LIVE framework to deal with the customized video editing task. We delicately devise both the training and inference pipeline to overcome the challenges of this task; 3) Extensive experiments demonstrate the effectiveness of our method in this task, yielding promising results in aspects such as image object swap, video consistency, motion maintenance, etc.

## 2 RELATED WORK

### 2.1 IMAGE GENERATION AND EDITING

While many deep generative models, such as GANs (Goodfellow et al., 2020), have showcased their capacity to generate realistic images (Brock et al., 2018; Karras et al., 2019), Denoising Diffusion Probabilistic Models(DDPMs) (Ho et al., 2020) have recently gained popularity due to their high-quality output on large-scale datasets (Dhariwal & Nichol, 2021). DALLE-2 (Ramesh et al., 2022) further enhances the text-image alignments utilizing the CLIP (Radford et al., 2021) feature space, while Imagen (Saharia et al., 2022) employs cascaded diffusion models (Ho et al., 2022b) for high-fidelity image generation. To boost training efficiency, Latent Diffusion Models (LDM) (Rombach et al., 2022) have been introduced to manage the generation process in the latent space.

Besides the great success made in image generation, text-to-image(T2I) diffusion models have also disrupted the dominance of previous state-of-the-art models in text-driven image editing (Richardson et al., 2021; Tov et al., 2021). Building upon a pre-trained T2I diffusion model, recent works (Couairon et al., 2022; Kawar et al., 2022; Tumanyan et al., 2022; Yang et al., 2022) have achieved remarkable performance in text-driven image editing. Prompt-to-Prompt (P2P) (Hertz et al., 2022), based on the crucial observation that the spatial layout and geometric information of generated images are preserved in the text-image cross-attention map, accomplishes fine-grained control of the spatial layout in the edited image by directly manipulating the cross-attention maps during the generation process. InstructPix2Pix (Brooks et al., 2022) and Paint-by-Example (Yang et al., 2022) enable characterized image editing with user-provided instructions. Textual Inversion (Gal et al., 2022), DreamBooth (Ruiz et al., 2022), and XTI (Voynov et al., 2023) learn special tokens for personalized

concepts and generate corresponding images. Null-text Inversion (Mokady et al., 2022) focuses on null-text optimization, facilitating real image editing.

## 2.2 VIDEO GENERATION AND EDITING

Compared to image generation, video generation presents a bigger challenge due to its higher-dimensional complexity and the scarcity of high-quality datasets. Video Diffusion Models (Ho et al., 2022c) have devised a novel architecture using a 3D U-Net with factorized spacetime attention (Ho et al., 2019) to generate temporally-coherent results. Imagen Video (Ho et al., 2022a) utilizes a cascaded Video Diffusion Model to achieve high-resolution video generation. However, these methods require paired text-video datasets and cost much to train. Recently, TAV (Wu et al., 2022) has innovatively transformed a T2I model into a T2V model and fine-tuned it to reconstruct the input video, enabling one-shot tuning video generation. Our model initialization is inspired by TAV.

T2I diffusion models have excelled in image editing, but their application in video editing remains underexplored. Text2Live (Bar-Tal et al., 2022) combines layered neural representations (Lu et al., 2020) with text guidance to demonstrate compelling video editing results. Dreamix (Molad et al., 2023) employs a pre-trained Imagen Video (Ho et al., 2022a) backbone to perform image-to-video and video-to-video editing, with additional ability to change motion. Gen-1 (Esser et al., 2023) trains models jointly on images and videos for tasks such as stylization and customization. However, these methods above require costly training. As a result, a collection of fine-tuning methods has been proposed. Video-P2P (Liu et al., 2023), built upon the image editing method P2P (Hertz et al., 2022), modifies the attention map corresponding to the text prompt and uses local blending to maintain consistency for the remaining parts of the attention heatmap. It also proposes decoupled-guidance attention control to enhance the preservation of the unedited area. Edit-A-Video (Shin et al., 2023) also uses attention replace mechanism in P2P and null-text inversion (Mokady et al., 2022) to edit videos and maintain coherence. Pix2Video (Ceylan et al., 2023) introduces self-attention feature injection, which can maintain edited videos' appearance coherence. It further develops latent guidance, using the l2 loss between the predicted current frame from the previous frame and the ground truth frame as guidance, significantly improving temporal consistency.

Although these methods have made decent progress in text-driven video editing, they are limited to video editing through modifications of the text prompt and cannot be applied to our proposed image-driven video editing task. As such, we build our framework on some modifications to the TAV framework and attention control methods, making it suitable for our task.

## 3 METHOD

In this section, we first briefly introduce DDPMs (Ho et al., 2020) in Sec.3.1. Then we introduce LIVE, a framework designed to overcome the image-driven video editing task. In Sec.3.2, we present our LIVE framework, which introduces a latent prompt designed specifically for completing the image-driven video editing task. Then we further explain how the framework generates edited videos in Sec.3.3 and attention control to achieve the best results in Sec.3.4.

## 3.1 PRELIMINARY

DDPM (Ho et al., 2020) is a recently popular generative model consisting of a forward noising process and a backward denoising process. The goal is to add noise to images step by step to Gaussian noise $z \sim N(0,1)$, and then to denoise images to obtain the restored images. It can be represented as:

$$q(x_t|x_{t-1}) := \mathcal{N}(x_t; \sqrt{1-\beta_t}x_{t-1}, \beta_t\mathbf{I}),$$
$$x_t := \sqrt{\bar{\alpha}_t}x_0 + \sqrt{1-\bar{\alpha}_t}\epsilon_t,$$

(1)

where $a_t := 1 - \beta_t$ and $\bar{\alpha}_t := \prod_{s=1}^{t} a_s$. The way to choose $\bar{\alpha}_t$ is called noise schedule. We choose cosine noise schedule as Nichol & Dhariwal (2021) mentioned. When $x_0$ is known, $q(x_{t-1}|x_t, x_0)$ is denotable from Eq.1 as $q(x_{t-1}|x_t, x_0) := \mathcal{N}(x_{t-1}; \frac{1}{\sqrt{\alpha_t}}(x_t - \frac{\beta_t}{\sqrt{1-\bar{\alpha}_t}}\epsilon_t), \frac{1-\bar{\alpha}_{t-1}}{1-\bar{\alpha}_t}\beta_t)$.

When it comes to the denoising process, it is natural to use $q(x_{t-1}|x_t)$ to tackle the backward process. Unfortunately, $q(x_{t-1}|x_t)$ is insoluble, so a deep neural network is used to learn the denoising

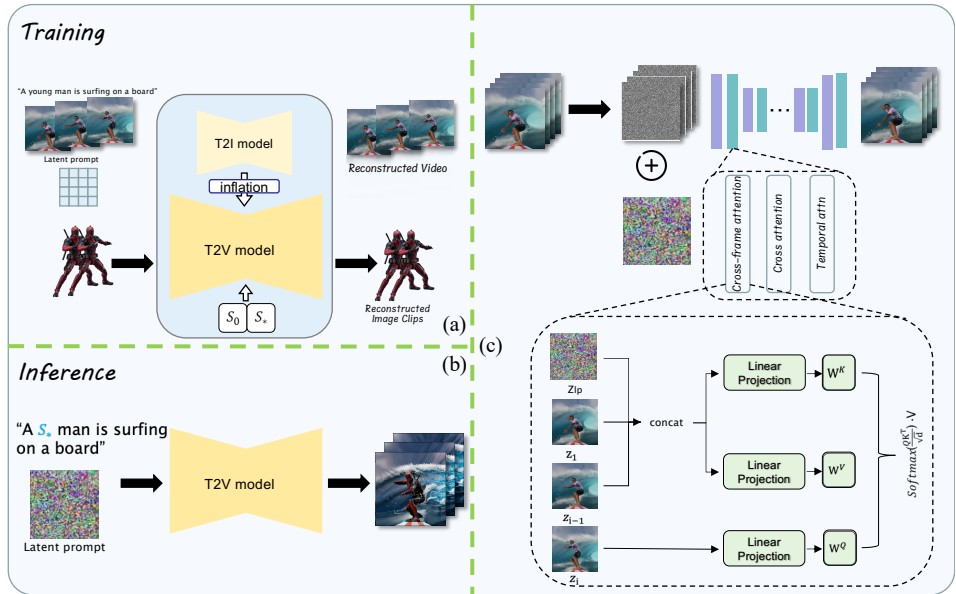

Figure 2: Illustration of the proposed LIVE framework. (a) Overview of *training:* We first inflate T2I models to T2V models, and train them with two objectives. The first one is to finetune on a text-video pair, getting trained T2V models. The second one is to align input images into text embedding, enabling characterized video editing abilities. (b) Overview of *Inference:* a modified text prompt is used to generate edited videos. (c) An extensive explanation of our *LIVE:* Our method takes input videos as the finetune objects. After the diffusion forward process gets noisy latents, we concat a latent prompt in frame dimension as unet input. We update attention blocks with diffusion reconstruction loss. As for the cross-frame attention blocks, features of latent prompt $z_{lp}$, first frame $z1$, and previous frame $z_{i-1}$ are projected to key $K$ and value $V$. Features of the current frame $z_i$ are projected to query $Q$. Outputs are sent to cross-frame attention blocks to update $z_i$.

distribution. It can be defined as follows:

$$p_\theta(x_{t-1}|x_t) := \mathcal{N}(x_{t-1}; \mu_\theta(x_t, t), \Sigma_\theta(x_t, t)), \tag{2}$$

where $\mu_\theta(x_t, t)$ can be represented by $\frac{1}{\sqrt{\alpha_t}}(x_t - \frac{\beta_t}{\sqrt{1-\bar{\alpha}t}}\epsilon_t)$. For the $\Sigma\theta(x_t, t))$ part, Nichol & Dhariwal (2021) finds that it has upper and lower bounds and set it to a constant to simplify the training objective as:

$$L_t = \|\epsilon - \epsilon_\theta(\sqrt{\bar{\alpha}_t}x_0 + \sqrt{1-\bar{\alpha}_t}\epsilon, t)\|^2. \tag{3}$$

However, it is known to all that DDPM's reverse process is stochastic and unstable, leading to different reconstructed results from inputs. Such behavior can be catastrophic for editing tasks since editing requires only partial changes within a specific region while maintaining the original appearance of the rest of the image. Fortunately, the introduction of DDIM (Song et al., 2020) has resolved this issue. During inference, DDIM sets $\Sigma\theta(x_t, t)$ to 0, which transforms the inference process from stochastic to deterministic. This leads to a denoising process that can be completed with fewer steps while ensuring that the denoised result is consistent with the input image, with only a modest reduction in quality. This denoising approach, also known as DDIM inversion, is integrated into our method.

## 3.2 LIVE FRAMEWORK OVERVIEW

The LIVE framework is depicted in Fig. 2(a) and (b). As shown in Fig. 2(a), the LIVE framework commences with a T2I model, which subsequently expands into a T2V model. In terms of technical implementation, we adopt the configuration of TAV (Wu et al., 2022), as it is one of the few publicly available T2V models. However, this model alone is insufficient for accomplishing image-driven video editing tasks due to its inability to generate coherent videos and establish a stable mapping with input images. Consequently, the first training objective focuses on acquiring the video reconstruction capability. During this phase, the inputs consist of the video to be fine-tuned and its

corresponding text prompt. After the video is encoded into latent representations by the Variational Autoencoder (VAE) and noise is added, we concatenate a learnable tensor ($1 \times 64 \times 64 \times 4$), referred to as the latent prompt, along the frame dimension. The shape of the latent prompt is identical to those of the video latents, except for the frame dimension. This concatenated tensor is then used as the input for the U-Net to obtain the reconstructed video. For the second training objective, it is necessary to incorporate the input image information into the T2V model. We make efforts to associate the input image with a specific space in the text embeddings. In such a way, the two objectives' training costs are at the same level. It facilitates alternating training and enables both training objects to simultaneously converge to their optimal. The final training objective is:

$$L_t = L_t^{unet} + \lambda L_t^{text}, \tag{4}$$

where $L_t^{unet}$ denotes diffusion loss for optimizing U-Net and $L_t^{text}$ denotes diffusion loss for optimizing text embedding. Their specific forms are the same as those presented in 3. $\lambda$ is the loss coefficient to balance two parts training effect, which is set to 1 in practice.

After the model has been trained, image-driven video editing can be accomplished through a straightforward process, as depicted in Fig. 2(b). First, we need to substitute the object name in the text prompt of the original video with the rare string $S_*$ and use it as input for inference. Then, during inference, we utilize attention control mechanism to obtain the final edited video.

The core of the LIVE framework lies in the latent prompt. The success of the image-driven video editing task hinges on accurate object replacement and the seamless motion flow of the replaced object. Consequently, the motion information learned by the model must be adequate to support the object's movements. The introduction of the latent prompt aims to resolve this issue by adding a learnable variable to compensate for the insufficient motion information in the inflated T2V model.

### 3.3 LATENT PROMPT BASED VIDEO EDITING

A natural approach would be to employ the TAV model with textual inversion to tackle this task. However, as demonstrated in Figure 3, when employing only cross-frame attention, the movements of the replaced character exhibit discontinuities, resulting in artifacts within the character's motion. This limitation arises because the TAV model operates as a T2V model, which has undergone only one-shot fine-tuning. The motion information learned by the model is insufficient to support the seamless motion alignment of image information generated through textual inversion. Hence, we start to rethink the attention mechanisms in TAV and propose the latent prompt to enhance models' motion perception.

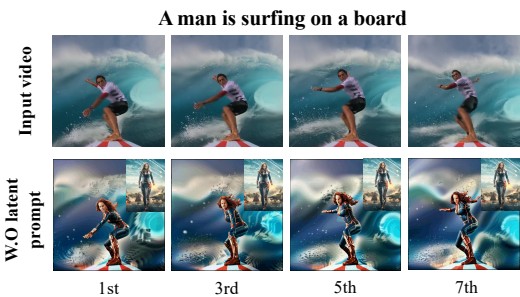

Figure 3: Observations on inflated T2V model's editing abilities. The top row is the finetuned video, and the second row explicitly uses an inflated T2V model to get edited video. It can be seen that hands movement is terrible and sometimes hands can disappear.

**Latent prompt.** To address the intricacies of video frame manipulation, we introduce the concept of the latent prompt that operates on two crucial dimensions: spatial and temporal. This spatial-temporal interaction is pivotal for achieving coherent and seamless video edits.

Given the video latent $z \in \mathbb{R}^{f \times h \times w \times c}$, the latent prompt $z_{lp}$ is defined as a latent in the same latent space as $z$ but only has one frame, where $z_{lp} \in \mathbb{R}^{1 \times h \times w \times c}$. We initialize the latent prompt with the first frame of the input video. The spatial interaction begins with the latent prompt performing cross-attention across all frames of the video. This operation enables $z_{lp}$ to grasp intricate spatial details and relationships between different parts of the video frames. By fusing spatial information from various frames, the Latent Prompt equips the model with a richer understanding of the visual content, allowing for more precise manipulation of the video frames. As for the temporal dimension, the Latent Prompt plays a vital role in learning and encoding the motion dynamics within the video sequence. With the aid of cross-attention mechanisms and temporal attention, the latent prompt has acquired a comprehensive understanding of the video's temporal dynamics.

**Training.** Starting with the first training objective, the latent prompt is a tensor in the latent space that can be optimized, similar to the text prompt mechanism used in the text encoder. As in Fig.2(c), the latent prompt is concatenated with input video latents along the frame dimension and fed into the U-Net for training. Owing to the addition of the latent prompt, we modify the attention mechanism by incorporating the latent prompt $z_{lp}$, the first frame $z_1$, and the previous frame $z_{i-1}$ as attention inputs, shown in Fig.2(c) downside. The formulation of our attention mechanism is as follows:

$$Q = W^Q z_i, K = W^K[z_{lp}, z_1, z_{i-1}], V = W^V[z_{lp}, z_1, z_{i-1}], \tag{5}$$

where $[\cdot]$ denotes concatenation operation. This ensures that the latent prompt maintains cross-frame attention with each frame in the sequence. Consequently, the latent prompt is enriched with video motion information, enhancing the naturalness and smoothness of the motion in videos generated by the T2V model, which is inflated from the T2I model.

Considering the second training objective, we begin with textual inversion (Gal et al., 2022) by providing rare strings, such as " $< aka* >$ ", with the initial token's embedding values for initialization, which aids in creating alignment between the rare strings and images. We use the original image searched from Internet as the input, without any segmentation. Due to the introduction of the latent prompt, it becomes necessary to incorporate a latent prompt into the input. As depicted in Fig. 2(a), our input is a combination of input image latents and their copies, with one representing the latent prompt and the other representing the standard input for textual inversion. Furthermore, we discover that the temporal attention in the U-Net architecture has an adverse impact on learning this component. Consequently, we opt to bypass the use of temporal attention during the training of the second objective to achieve optimal training results.

To achieve a balance between performance and training time, our approach involves updating the latent prompt concurrently with the U-Net architecture. However, during the second training objective, we freeze the latent prompt and replace it with latents of the input image to satisfy the requirements of the second part of the training, which largely preserves the textual inversion training results. Initially, we employ a cross-attention mechanism that takes both text and video as input and outputs the latent prompt for training. However, we discover that this significantly more complex approach does not produce satisfactory results, potentially due to the limited training steps. As our goal is to fine-tune the model, we opt for a learnable variable instead of the previous complex mechanism, which leads to better results and improves training efficiency.

**Inference.** After completing the training, we can directly obtain the edited video from the rare string used during training. However, we need to analyze the processing of the latent prompt. During the training of the first objective, we train the latent prompt using a trainable vector, while in the second part, we replace the latent prompt position with the image latents. When inferring diffusion model, it usually adopts the classifier-free guidance (Ho & Salimans, 2022) method. As for the conditional part, we set the input as the trained latent prompt and noise to enhance the motion information required for the input images. When it comes to the unconditional part, we set the input as image latents and noise so that the generated videos can include not only the image information bound by the text prompt through the second part of the LIVE framework but also supplement it with cross-frame attention, which can create videos that are content-rich and temporally continuous.

## 3.4 ATTENTION CONTROL

After training a T2V model with the LIVE framework, we can perform object swapping. Attention control has gained widespread attention as a training-free editing method. It guides the model to generate edited images while preserving specific information from the original image by swapping the maps of self-attention and cross-attention layers. We extend this concept to the video level to accomplish object replacement while maintaining consistency between the original video and the edited version in areas without objects. Following the setting of P2P (Hertz et al., 2022), after getting origin prompt attention mask $M_t$ and edited prompt attention mask $M_t^*$ at timestep $t$, we adopt Word Swap edit function $Edit(M_t, M_t^*, t)$, which is expressed as follow:

$$Edit\,(M_t, M_t^*, t) := \begin{cases} M_t^* & \text{if } t < \tau \\ M_t & \text{otherwise} \end{cases}, \tag{6}$$

where $\tau$ denotes the starting of attention injection. Due to the observation that early steps of diffusion models contribute to the overall distribution of results, we inject edited attention mask $M_t^*$

**[Input Video]** a man is driving a motorbike in the forest

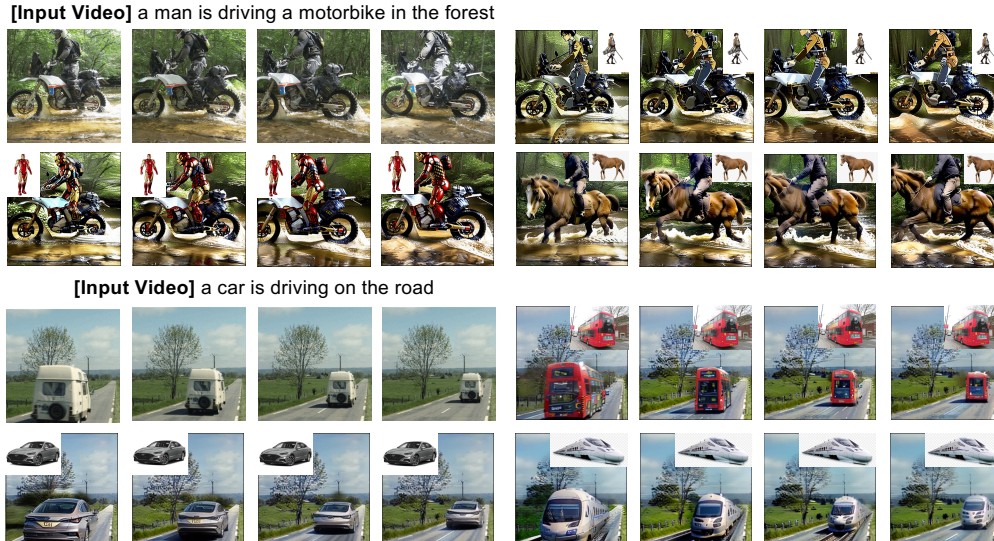

**[Input Video]** a car is driving on the road

Figure 4: Videos editing results from various input videos, images, and prompts. Our LIVE framework can accurately replace the designated object in the input image with the corresponding object in the video while maintaining the visual continuity of the edited video.

to guide the composition generation. The original Word Swap can only be applied to text prompts of the same length and replace them with complete attention. We modify it to specify which part of the text prompt needs replacement so that we can have a flexible plan for Word Swap. We typically select verbs that represent motion and nouns that represent background in the sentence for replacement. We did not choose to replace the object because we need the edited video to contain accurate objects from the input images. We do not want the object information in the origin attention map to interfere with our editing. We replace the background to ensure that the generated object matches the new background. However, we have discovered that only replacing the background is not sufficient, because the background's attention map is not as accurate as the objects'. VideoP2P (Liu et al., 2023) and other text-driven video editing methods (Shin et al., 2023) utilize null-text inversion (Mokady et al., 2022) to tackle this problem, but these approaches increase the model training cost. Therefore, we choose to use local blend, another method in P2P (Hertz et al., 2022), to compulsively replace the background and then apply Gaussian blur to adjust the transition between objects and the background, which also yields promising results.

## 4 EXPERIMENTS

**Implementation.** We implement our approach based on stable diffusion v1-4[1]. Regarding optimization of U-Net, similar to TAV (Wu et al., 2022), we fine-tune only the attention modules inside the U-Net architecture on 8-frame $512 \times 512$ videos with a learning rate of 3e-5 for 500-600 steps. For the textual inversion component, we use only one single input image with the placeholder token set to " $< @\#\$ >$ " and train it using a learning rate of 5e-03 for 500-600 steps. As for the latent prompt, we initialize it with the first frame of the original video during the first part of training and fix them to latents of the input image during the second part of training. During inference, we start from the saved DDIM inversion values (Song et al., 2020) and set the classifier-free guidance (Ho & Salimans, 2022) to 12.5. We use DDIM sampler to infer results for 50 steps. For attention control, we set the cross-attention replacing ratio to 0.6 and the self-attention replacing ratio to 0.3. Experiments are conducted on an RTX 3090ti GPU, with 15 minutes for training 500 steps and 10 minutes for getting DDIM inversion values and inferring.

**Baselines.** As our proposed task has not been explored before, we choose the concurrent text-driven video editing method as a baseline for comparison. The baseline methods include: (1)*TAV+DDIM (Wu et al., 2022)*: This method generates videos from text prompts by fine-tuning pre-trained T2I diffusion models using an efficient one-shot tuning approach. (2)*Video-P2P (Liu et al., 2023)*: Building upon TAV, this method enables control over video content through text editing using attention control and null-text inversion. (3)*Vid2Vid-zero (Wang et al., 2023)*: This

---

[1]Stable Diffusion: `https://huggingface.co/CompVis/stable-diffusion-v1-4`

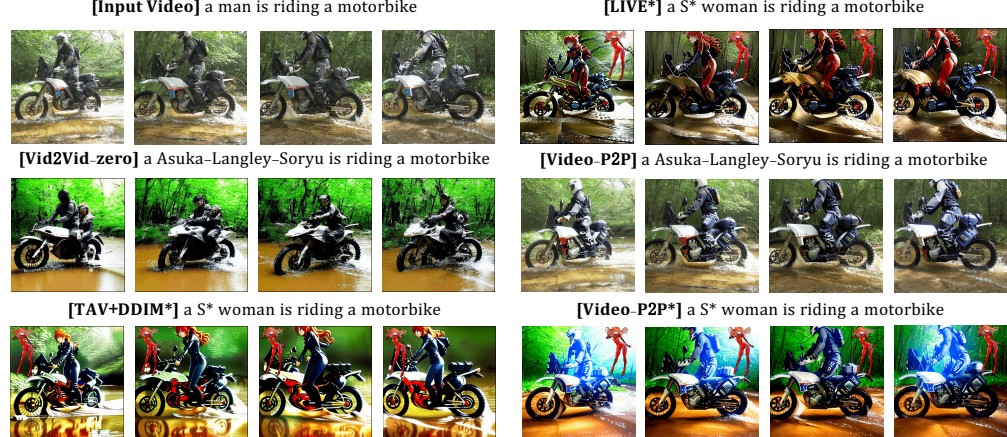

Figure 5: Comparison to baseline methods. ∗ means that this method is implemented with textual inversion. Only LIVE achieves both accurate object replacement and video consistency.

approach achieves zero-shot video reconstruction through dense spatio-temporal attention modules, and realizes video editing with attention control and null-text inversion. Experiments are conducted with their official codes and configurations

**Dataset.** We use the conventional videos used in Video-P2P (Liu et al., 2023) and Vid2Vid-zero (Wang et al., 2023) as the evaluation dataset. We use their default text prompts to edit videos for fair comparison.

### 4.1 MAIN RESULTS

**Qualitative results.** We provide a set of examples of our method implementation in Fig. 4. For each example, we show the 2nd, 4th, 6th, and 8th frames of the input and edited videos, as well as the corresponding text prompt and input images. As demonstrated in the figure, LIVE is capable of handling videos with prominent foreground objects and multiple foreground objects. We can accurately replace objects with the input image, and the motion of the objects appears natural and the same as the original video. Furthermore, our method can handle a wide range of object replacements. For example, as demonstrated in the second row, we can replace a motorcycle being ridden by a person with a horse, or as shown in the fourth row, we can replace a car with a train. These examples demonstrate the effectiveness of our approach. Please note that the color issues in the background of the fifth and sixth rows are caused by limitations in the TAV approach used for reconstructing the original video, and are not related to the LIVE framework itself.

**Comparisons.** From our comparison with baselines, all of them are designed to address text-driven video editing. Except for Vid2Vid-zero, we implement textual inversion in other baselines, enabling them to generate input objects. Vid2Vid-zero is a training-free method, while textual inversion needs to optimize text embedding, making it not suitable for adding textual inversion. However, it can be observed from Fig.5 that their performance in image-driven video editing is not very satisfactory. Vid2Vid-zero can generate general results but poor motion. Video-P2P without textual inversion can generate much better results, with impressive preservation of background. But the motion and pose of the object are blurry. TAV+DDIM performs the best among the three baselines by effectively replacing the object in the input image into the video. However, the edited video may struggle to maintain the original content of the video in the background. As shown in Fig. 5 above, our approach successfully achieves both object replacement and background preservation. The generated video using our approach exhibits smoother and more continuous motion, resulting in better overall video coherence.

**Quantitive results.** 1) *CLIP score:* We follow the setting of (Ho et al., 2022a) to calculate CLIP score (Park et al., 2021), evaluating text-video alignment.We calculate CLIP scores at the frame level and take average results as the final score. As shown in Fig. 6(a), our LIVE outperforms any baseline method, demonstrating that our method possesses better text-video alignment. 2) *User study:* We further evaluate LIVE against baselines with a user study. The questionnaire includes 5 videos, each

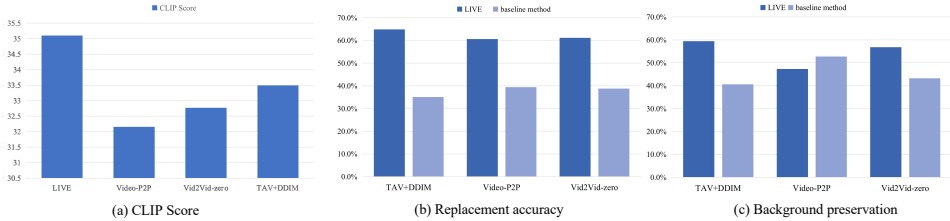

Figure 6: Quantitative results. The user study shows that our LIVE framework gets better performance in object replacement accuracy, background preservation, and clip score than other baselines.

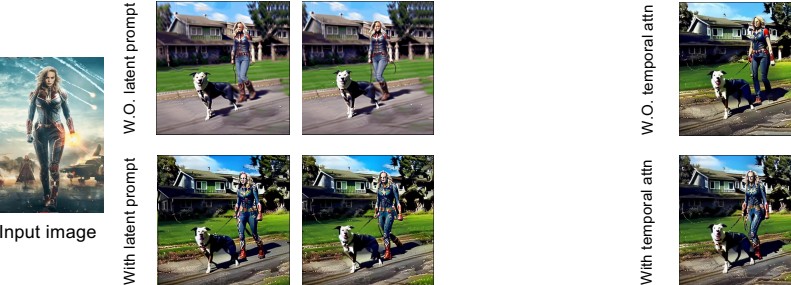

Figure 7: Ablations on latent prompts.    Figure 8: Ablations on temporal attention.

of which is edited using four methods, including LIVE. For each video, there are two questions: 1) Which method best preserves the input image after editing? 2) Which method has the best-preserved background after editing? These two questions correspond to the two challenges of our proposed image-driven video editing task: accurate object replacement and video coherence. In Fig.6(b) and (c), the majority of the participants agree that our method can more accurately replace objects than any other baseline methods. As for background preservation, we beat most methods except for Video-P2P (Liu et al., 2023).Video-P2P (Liu et al., 2023) utilizes null-text inversion (Mokady et al., 2022) and delicately devises an attention control mechanism to achieve slightly better preservation results. In summary, the user study confirms that our method performs much better than other methods in object replacement.

## 4.2 ABLATION STUDIES

**Latent prompt.** Our primary focus is on the latent prompt. Thus, we initially examine its influence on the overall framework. As shown in Fig. 7, the top row shows the case without latent prompt, and it can be seen that the texture and shape of the object cannot be well preserved, and strange motion can occur. In the next row, after adding the latent prompt, precise object replacement is achieved, and the motion is also continuous, without the strange case in the top row.

**Training of temporal attention.** In the second part of the training, the goal is to bind text and the input image, which theoretically should be trained on a 2D model without the time dimension. However, since we inflate the model to 3D, temporal attention is involved during training. Intuitively speaking, temporal attention should not be trained in the second part, as it would affect not only temporal attention but also the quality of the binding between text and images. As shown in Fig. 8, when temporal attention is trained together with the second part, the image learning is not satisfactory and artifacts may appear. However, freezing temporal attention during the second part of training leads to a significant improvement in overall results.

## 5 CONCLUSION

In this paper, we introduce a novel video editing scenario called image-driven video editing, which aims to replace objects in videos with input images. To address this task, we propose a novel LIVE framework in this paper. By utilizing the proposed LIVE, users can precisely control the editing process and achieve impressive performance on real-world images, which provides a flexible way for content modification and creation. We hope that this work can serve as a solid baseline for future research in image-driven video editing and also become an essential support for future work.

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

## A   APPENDIX

You may include other additional sections here.

