# OpenReview forum: "Image-driven Video Editing with Latent Diffusion Models"
_ICLR.cc/2024/Conference — ICLR 2024 Conference Withdrawn Submission_

### Official Review · Reviewer_PX4c · 2023-10-30

**Soundness:** 3 good
**Presentation:** 3 good
**Contribution:** 2 fair
**Rating:** 5
**Confidence:** 3

**Summary:**

The paper focuses on the image-driven editing task, and proposes a framework (LIVE) by inflating a T2I model to T2V model, introducing a new Latent Prompt code, and an attention control way.
During the training, a learnable latent prompt code is also learned. The learned latent prompt code later is attended together with the frame latent codes in the attention layers during the inference.
LIVE achieves better editing fidelity and relatively fair temporal consistency compared to other baseline methods.

**Strengths:**

General idea:
- The paper proposes to use a reference image to edit a video. This seems to be able to work as task to customize the video.

Method:
- The introduction of Latent Prompt with the inconsistency from T2V model makes sense to me.
- The main idea of using Latent Prompt and replacing the subject with learned prompt code is interesting.

Results:
- LIVE indeed shows better performance in terms of the object replacement.

Writing:
- The paper is well-written and easy to follow.

**Weaknesses:**

Experiments & results:
- Quality. The quality of results is generally not satisfactory, especially in terms of the consistency.
- The ablation study seems not to be sufficient. For example, authors also mention they did not replace the objects as they wanted to keep the accurate object in the edited video, but this does not appear in the ablation study.
- The number of qualitative results. Although LIVE seems to work better on the conventional videos in Video-P2P and Vid2Vid-zero, it is hard to evaluate it because the size of the dataset is too small. The evaluation itself has some limitations and does not show the full performance of the proposed method.
- Video scenarios. The proposed method is mostly tested on the videos that has motion horizontally. How does the method work for the videos that have more complicated motion? For example, a person turning around or an animal jumping?
- Limitations. The paper does not include any failure cases, which is quite common in image/video synthesis papers. Having some limitations can help people better understand the proposed method from different perspectives.

Writing:
- Some typos.
   - In Figure 2 caption, "unet"-> "U-Net", $z1$ -> $z_1$.

**Questions:**

Method:
- In Figure 8, what is the artifact with training temporal attention?

**Details Of Ethics Concerns:**

The paper introduces a method that could be used to generate videos with specific subject, which could be an issue for misinformation or the abuse of the copyright.

---

### Official Review · Reviewer_W3nE · 2023-10-31

**Soundness:** 2 fair
**Presentation:** 2 fair
**Contribution:** 3 good
**Rating:** 3
**Confidence:** 5

**Summary:**

This paper provides an effective solution for image-driven video editing. Specifically, the authors design a novel image-driven video editing task aimed at generating customized videos based on the input image. Numerous experiments have proven the effectiveness of the proposed method. The manuscript is technically sound, and the manuscript is clearly written overall with helpful schematic illustrations and, in particular, a good survey of related work.

**Strengths:**

The core contributions of this paper are listed as follows:
1 The authors introduce a novel image-driven video editing task aimed at generating customized videos based on the input image.
2 The authors propose a LIVE framework to deal with the customized video editing task. We delicately devise both the training and inference pipeline to overcome the challenges of this task.
3 Extensive experiments demonstrate the effectiveness of the proposed method in this task, yielding promising results in aspects such as image object swap, video consistency, motion maintenance, etc.

**Weaknesses:**

1. Didn't see a reference to Figure 1 in the paper. Please introduce Figure 1 in the paper and further explain what the S* in the figure means.
2. The exact meaning of the different parts of Figure 2 is not clear. In a, are the text-video pair and input images given as input together, or are they split into two training sessions? In b, the input for inference should contain three parts: the original video, the input prompt, and the reference image, why is there only an input prompt here? In c, please give the meaning of each part in words.
3. The paper uses a video length of 8 frames for the experiments, but does not provide a detailed analysis of the impact of the video length on the performance of the proposed method.
4. The paper briefly mentions that the proposed method is more efficient than some other methods, but does not provide a detailed analysis of the computational efficiency. This may be an important factor for practical applications.
5. The paper does not provide a detailed discussion of the limitations of the proposed method, such as the types of videos that may not work well with the method or the potential failure cases.
6. What is the source video input in Figure 7 and Figure 8?

**Questions:**

1. In Figure 1, what is the BASELINE method for text-driven video editing? Is it TAV?
2. What is the meaning of S0 and S* in Figure 2?
3. Are the prompts in Figure 4 always the same? For the last four graphs in Figure 4, what are the input prompts? Is it ‘a car’ or ‘a train’?
4. The paper mentions using TAV as a baseline, so why is there a need to inflate T2I as T2V?

---

### Official Review · Reviewer_oLny · 2023-11-01

**Soundness:** 2 fair
**Presentation:** 2 fair
**Contribution:** 2 fair
**Rating:** 5
**Confidence:** 4

**Summary:**

This paper presents a latent diffusion-based method to edit a video based on a joint conditioning of input video, textual description and image. In the edited video, the subject that is associated with a rare token in the text prompt will reflect the style from the input image. This is done by expanding a text-to-image diffusion model and training it with an input video to capture the motion of the input video and training a token embedding using textual inversion to associate the input text prompt with the conditioning image. The model additionally trained a latent prompt to refine the edited video.

**Strengths:**

1. The visual results presented in the demo are appealing.
2. The method uses a single conditioning image but the output captures the style in the conditioning image well even if the viewing angle of the edited video is different from the conditioning image.

**Weaknesses:**

1. The writing of the paper is poor. Section 3 Methodology should be restructured so that each training objective is expressed clearly
2. More experiments and ablation studies are needed to justify the effectiveness of each component in the model.
3. In Section 3 equation (4), the loss function combines the loss for training the video diffusion model and the loss for training the text embeddings in textual inversion. It seems like with this joint training objective the two parts are trained together. However, in section 3.3 the author mentioned "However, during the second training objective, we freeze the latent prompt..." which sounds like the training is conducted in two phases. It is unclear to me how the training is conducted with the two training objectives.
4. For the TAV+DDIM baseline model, how did the author enable image conditioning with this method? As far as I know, TAV takes text and video inputs and does not explicitly condition on input images. Did the author use a personalized dreambooth model as the backbone of TAV to condition on the input images?
5. Based on the demo it seems that LIVE works well with a single conditioning image while textual inversion and dreambooth require at least 3-5 input images to train a personalized model. It would be interesting to discuss why LIVE worked well with only 1 image.

**Questions:**

Please refer to the weakness part.

---

### Official Review · Reviewer_9nKT · 2023-11-01

**Soundness:** 3 good
**Presentation:** 2 fair
**Contribution:** 3 good
**Rating:** 3
**Confidence:** 3

**Summary:**

The paper addresses the challenge of image-driven video editing, specifically swapping an object in a video with another object from a given image.

It introduces a novel framework, LIVE, which fine-tunes a Text-to-Image latent diffusion model and learns additional latent parameters tailored to each (video, caption, image) sample. This approach, coupled with the prompt-to-prompt technique, effectively enables object swapping in videos.

Assessment on a modest-sized video dataset, previously utilized by other researchers, demonstrates that LIVE outperforms various baseline methods in terms of editing quality, as measured by CLIP score, replacement accuracy, and background preservation.

**Strengths:**

The paper introduces the task of image-driven video object swapping, presents a plausible framework, and delivers promising results.

**Originality**: The concept of swapping objects in videos via images is innovative. Although the framework primarily integrates existing components, its application is sufficiently novel.

**Quality**: The framework's high-level rationale is sound, and it outperforms several baselines. However, better supporting its claims could further enhance the paper's quality, as discussed in the weaknesses section.

**Clarity**: The paper's clarity is lacking. Ambiguous and obscure sections impede comprehension, especially concerning technical details. We will elaborate on this in the weaknesses and questions sections.

**Significance**: As an inaugural approach to image-driven video object swapping, the paper has the potential for great impact, inspiring further research and real-world applications.

**Weaknesses:**

The paper claims three contributions: the novel task, the LIVE framework, and extensive experiments. However, these claims lack robust substantiation.

1. **Task Definition**: The paper fails to formally define the task. It is unclear whether a video caption is essential or merely a requirement for the proposed framework. The scope of video editing covered (e.g., object swapping, addition, removal, style changes) remains ambiguous.

2. **Framework Choices**: The paper does not explore alternative options for each framework component or their respective advantages and drawbacks. For instance, it does not justify the decision to learn a token embedding without also fine-tuning the text encoder, as done in methods like DreamBooth.

3. **Experimental Rigor**: Labeling the experiments as "extensive" is misleading given the limited number of test videos. Although the method outperforms baselines, notable artifacts remain (e.g., over-exposure, unintended changes in the background or other objects). A more systematic evaluation is necessary.

The method description could also be confusing or vague in several aspects:

4. How to actually "inflate" a T2I model to a T2V model (compared with the inflation proposed in the Tune-A-Video paper)? In Eq (5), it should be the intermediate feature maps rather than the input latent codes $z_*$, right? Furthermore, if we concatenate the features, the dimension of the matrix $W^K$ and $W^V$ should be expanded as well?

5. In Eq (4), the proposed method optimizes $L_t^{unet}$ first and then optimizes $L_t^{text}$. Why would it be equal to optimizing the sum of them? It should also rigorously define the loss terms (e.g., what are the actual learnable parameters?).

6. The necessity of the prompt-to-prompt technique, especially the replacement of verbs and the background, is not convincingly argued. Why would only replacing the subject not sufficient? What would the attention maps look like?

Moreover, practical concerns are raised:

7. How can we determine when to stop the fine-tuning? I assume that we do not want to "over optimize" the loss terms to zero (which could destroy the original T2I model's capability), right?

8. The lengthy processing time (25 minutes for an 8-frame video?) could significantly hinder the framework's practical impact. A discussion on the trade-off between quality and latency is needed.

**Questions:**

I would appreciate if the authors can address the weaknesses section above.